# Immunological Aspects of Cancer Cell Metabolism

**DOI:** 10.3390/ijms25105288

**Published:** 2024-05-13

**Authors:** Sisca Ucche, Yoshihiro Hayakawa

**Affiliations:** 1Institute of Natural Medicine, University of Toyama, Toyama 930-0194, Japan; sisca.ucche@mail.ugm.ac.id; 2Faculty of Pharmacy, Universitas Gadjah Mada, Yogyakarta 55281, Indonesia

**Keywords:** cancer, progression, metabolism, immune surveillance, immune escape

## Abstract

Cancer cells adeptly manipulate their metabolic processes to evade immune detection, a phenomenon intensifying the complexity of cancer progression and therapy. This review delves into the critical role of cancer cell metabolism in the immune-editing landscape, highlighting how metabolic reprogramming facilitates tumor cells to thrive despite immune surveillance pressures. We explore the dynamic interactions within the tumor microenvironment (TME), where cancer cells not only accelerate their glucose and amino acid metabolism but also induce an immunosuppressive state that hampers effective immune response. Recent findings underscore the metabolic competition between tumor and immune cells, particularly focusing on how this interaction influences the efficacy of emerging immunotherapies. By integrating cutting-edge research on the metabolic pathways of cancer cells, such as the Warburg effect and glutamine addiction, we shed light on potential therapeutic targets. The review proposes that disrupting these metabolic pathways could enhance the response to immunotherapy, offering a dual-pronged strategy to combat tumor growth and immune evasion.

## 1. Background

Normal healthy cells can evolve into transformed cancer cells through various mechanisms, including genetic mutations that lead to uncontrolled cell growth, epigenetic alterations that change gene expression without altering the DNA sequence, and disruptions in cellular signaling pathways that govern cell proliferation and death [1]. During such cancer development, it is known that immune cells are essential for controlling the development and progression of cancer cells. This process has been regarded as a cancer immune surveillance process where immune cells work cooperatively to eliminate cancer cells to protect the host. However, an extensive series of previous works revealed that the host immune system works not only for protection but also for scalping or editing cancer cells to escape the immune surveillance process, thereby being revised as a cancer immune-editing process. Through the cancer immune-editing process, cancer cells proceed to genetic and epigenetic changes that form a new tumor variant under the potent selection pressure from immune cells. Alternatively, immune cells from a heterogeneous cancer population can select cancer cells with poorer immunogenicity and/or immune-evading properties. Those immune-escaped cancer cell variants can proceed with the disease progression to proliferate and metastasize [2].

Considering the plasticity nature of cancer cells, several mechanisms of cancer cells for immune escape have been noted, such as decreasing their immunogenicity and/or inducing an immunosuppressive environment [3]. The tumor microenvironment (TME) plays a crucial role in regulating the growth and progression of malignant tumor cells. The TME comprises various types of cells, including cancer, endothelial, fibroblasts, and immune cells, all contributing to cancer cell growth and invasion [4]. In turn, cancer cells constantly modify the TME to escape from immune surveillance [5]. Interestingly, the metabolic competition between tumor and immune cells has also been suggested as a potential mechanism for cancer cells to escape from the immune surveillance process [6]. In general, the metabolic status of cancer cells is dysregulated by enhancing glucose consumption to support their anabolic requirements. There has been increasing evidence that cancer cell metabolism would impact the immunological status of TME and lead to their escape from immune surveillance and/or resistance to immunotherapy (Figure 1). In this review, we will discuss the potential role of cancer cell metabolism in the context of the immune response against cancer cells.

## 2. Immunological Components of Tumor Microenvironment

TME is a complex and dynamic ecosystem with many cellular players, including malignant and non-tumor cells, such as immune cells and stromal components [4]. Understanding the immunological landscape within the TME is crucial for grasping how tumors manipulate immune responses to evade detection and destruction. The TME is populated by diverse immune cells—T lymphocytes, macrophages, natural killer (NK) cells, and dendritic cells, each playing distinct roles. T lymphocytes, particularly cytotoxic T cells, strive to attack and destroy tumor cells; however, their efficacy can be severely diminished due to the immunosuppressive environment fostered by the tumor. Macrophages within the tumor, often called tumor-associated macrophages (TAMs), typically exhibit a phenotype that promotes tumor growth and immune suppression. NK cells offer the potential for rapid and direct responses against tumor cells, but can find their activity curtailed by various tumor-derived inhibitory signals. Dendritic cells, pivotal in antigen presentation, are frequently manipulated by cancer cells to prevent the effective priming of T cells, thereby dampening the immune response.

Adjacently to these immunological components, the metabolism of non-tumor cells in the TME plays a significant role in cancer progression and response to treatment. Non-tumor cells such as cancer-associated fibroblasts (CAFs) and endothelial cells undergo metabolic reprogramming to adapt to the metabolic demands of the TME. CAFs, for example, adopt a glycolytic metabolism to survive in the low-oxygen conditions often present within tumors, supporting cancer cells through the secretion of growth factors and modification of the extracellular matrix. Immune cells within the TME also exhibit metabolic flexibility: their metabolic pathways are intricately linked to their activation states and functions. For instance, effector T cells ramp up glycolysis to fulfill the energy demands for rapid proliferation and effector functions. In contrast, regulatory T cells might rely more on oxidative phosphorylation, reflecting their different roles and energy needs.

By exploring the immunological components and the metabolic adjustments of non-tumor cells in the TME, we can uncover new therapeutic targets and develop strategies that disrupt the supportive environment tumors need to grow. This dual focus not only enhances our understanding of the biological underpinnings of tumor progression but also opens up avenues for novel interventions aimed at reprogramming the TME to reinstate effective immune surveillance and suppression of tumor growth.

## 3. Metabolic Changes of Cancer Cells

The metabolic regulation between normal and tumor cells is known to be different. In normal cells, glucose is predominantly oxidized via mitochondrial oxidative phosphorylation (OXPHOS) to efficiently produce adenosine triphosphate (ATP) when oxygen is sufficient. However, under hypoxic conditions, glucose is fermented to lactic acid. In contrast, cancer cells can exhibit a different metabolic behavior known as the Warburg effect, preferentially fermenting glucose to lactic acid even in ample oxygen [7]. This shift is not solely driven by oxygen scarcity, but is a hallmark of cancer metabolism, supporting rapid proliferation and various survival strategies.

Cancer cells rapidly produce more ATP than the conventional OXPHOS pathway through glycolytic metabolic reprogramming. Moreover, abundant intermediate products of glycolysis can be used in the biosynthetic pathway and synthesis of macromolecules to support tumor growth. Generally, glucose can be taken up by glucose transporter (GLUT) family members, and modulation of GLUT is known to be associated with advanced tumor growth and invasion in various tumor cells [8]. Furthermore, the deficiency in GLUT1 enhances CTL-mediated tumor cell destruction. Inactivation of GLUT1 shifts the metabolic pathway towards oxidative phosphorylation, resulting in increased production of reactive oxygen species, enhancing TNF-α-mediated cell death dependent on caspase 8 and Fadd signaling pathways. The study further demonstrates that genetic and pharmacological inhibition of GLUT1 enhances the efficacy of antitumor immunity and improves the performance of anti-PD-1 therapy in mouse models, suggesting a potential therapeutic avenue for improving immune checkpoint blockade therapies [9].

In addition, cancer cells often upregulate the expression of the hexokinase (HK) enzyme family. HK is the first regulatory glycolysis enzyme that phosphorylates glucose into glucose-6-phosphate (G6P). Among the HK family, HK2 is selectively overexpressed in cancer cells. A high expression of HK2 has been observed in the majority of tumors, and this increase in HK2 levels has been correlated with the progression of tumors. Additionally, HK2 expression has been related to immune cell infiltration, such as T follicular helper cells (Tfh) [10].

In addition to glucose, amino acids are essential for the proliferation and survival of cancer cells, and glutamine is the most common amino acid that is known to be associated with cancer progression. Glutamine supports the proliferation of tumor cells by providing metabolic intermediates as a source of carbon and nitrogen to maintain redox equilibrium against oxidative stress [11]. Glutamine is transported into the cytoplasm with the help of transporters such as SLC1A5. Once glutamine enters the cytoplasm, glutamine is converted into glutamate by glutaminase (GLS) and further converted into a-ketoglutarate in the TCA cycle or used to generate glutathione for the antioxidant response. Tumor cells increase glutamine uptake by modulating the expression of glutamine transporter from the SLC family transporter. High expression of SLC1A5 has been found in many tumor cells and is related to tumor growth proliferation and survival [12].

Moreover, proliferating cancer cells also require many lipids to build their membrane during carcinogenesis [13,14]. Lipid is essential for maintaining the structure of the cells but also critical for providing energy and maintaining cellular signaling. Cancer cells often increase de novo synthesis of fatty acids (FAs) and lipid droplet metabolism [15,16]. An increase in fatty acid synthase (FASN), a key enzyme of de novo lipogenesis, has been reported to promote tumor growth and cell proliferation [17]. Furthermore, lipid peroxidation mediated through the CD36 receptor has been reported to impair CD8+ T cells within tumors. The study highlights how CD8+ T cells in the tumor microenvironment are impaired by the uptake of oxidized lipids through the CD36 receptor, leading to lipid peroxidation and functional decline. This pathway is crucial, as it contributes to the reduced efficacy of immunotherapies targeting these cells [18].

Although metabolic reprogramming in cancer cells represents a necessary cellular process for their malignant progression, the exact mechanism is how healthy normal cells acquire such metabolic features of cancer cells during the oncogenic process. A recent report revealed that the IFN-dependent immune-editing process guides cancer cells to engage in glycolytic metabolic programs and supports their proliferation and immune evasion [19]. This research offers a thorough comprehension of a complex process by which IFNγ triggers a non-canonical signaling cascade that involves STAT3 and c-Myc, leading to the manifestation of a unique metabolic phenotype of cancer cells. Through this IFN-γ-induced STAT3/c-Myc-dependent metabolic switch, cancer cells acquire a metabolic profile distinguished by increased aerobic glycolysis and decreased fatty acid oxidation. The increased expression of c-Myc, which is driven by IFN-γ through STAT-3-dependent signaling, has a crucial function in orchestrating a metabolic shift that not only supports tumor proliferation but also reduces the immunogenicity of the tumor, allowing the cancer to avoid detection by the immune system effectively in this model. While the role of IFN-γ in promoting antitumor immunity has been proposed elsewhere [20,21,22,23], its contribution to the metabolic reprogramming of tumor cells reveals a previously unexplored component of its biological function.

Furthermore, the involvement of a non-canonical signaling pathway that triggers STAT3 signaling allows IFN-γ to deviate from its conventional pathway of transcriptional activation mediated by STAT1 [24]. This finding underlines a currently overlooked adaptability in the dynamics of cytokine signaling. The significance of c-Myc in this particular circumstance in this research is remarkable. The upregulation of c-Myc activity, facilitated by the IFN-γ-STAT3 signaling axis, underscores the pivotal role of transcription factors in integrating signaling cues from the tumor microenvironment with cellular metabolism and growth. Moreover, c-Myc has been recognized as a critical regulatory molecule for controlling cell cycle and cell death [25]. In addition, c-Myc is also involved in metabolic reprogramming towards glycolysis; therefore, it underlines its multifaceted role in tumor progression and immune evasion. This observation is consistent with previous works that have demonstrated c-Myc to be a fundamental regulator of cellular metabolism to orchestrate a broad array of metabolic processes critical for the biosynthetic and energy demands of rapidly proliferating cells [26]. This study significantly advances our understanding of tumor immune evasion mechanisms, placing metabolic reprogramming at the forefront of such a complex process. Furthermore, these findings enrich the comprehension of cancer biology and establish a foundation for advancing innovative therapeutic strategies that capitalize on the metabolic susceptibilities of malignant cells by investigating the functions of IFN-γ, STAT3, and c-Myc in facilitating this reprogramming process. Thus, this work opened a new aspect of the immune-editing process: cancer cells can adjust their metabolic preferences to survive under the pressure of host antitumor immunity.

In summary, metabolic reprogramming in cancer progression and immune evasion offers insights into potential therapeutic strategies that target metabolic vulnerabilities in cancer cells.

## 4. Metabolic Molecules in Cancer Cells Associated with Immunosuppressive TME

In general, cancer cells with high glycolytic activity produce abundant lactate, and such increased lactate production of cancer cells decreases pH levels in the TME. Under low-pH conditions, antitumor immunity is known to be suppressed [27]. High acidification of TME was reported to regulate macrophage polarization and support tumor growth [28]. Lactate is also reported to directly regulate the polarization of macrophages by activating ERK/STAT3 signaling and promoting the progression of breast cancer [29], as well as the induction of myeloid-derived suppressor cells (MDSCs) by activating HIF-1α [30].

Indoleamine 2,3-dioxygenase (IDO), the rate-limiting enzyme that metabolizes tryptophan into kynurenines, is known to be involved in the immune escape of cancer cells [31]. IDO has been expressed in various cancer cells and is known to induce immunosuppressive TME [32]. IDO expression in brain tumors has been reported to inhibit the effector T cell function but increase the recruitment of immunosuppressive regulatory T (Treg) cells [33]. Furthermore, it has been reported that the overexpression of IDO in melanoma contributes to tumor resistance to T cell immunotherapies through the activation of MDSCs in a Treg-dependent manner [34].

Arginase is the key urea cycle enzyme that hydrolyses L-arginine to urea and L-ornithine to remove toxic ammonia [35]. Increased arginase expression has been found in several types of cancer and is associated with the induction of immunosuppressive cells to promote tumor progression [36,37,38]. It has been reported that high arginase 1 (ARG1) expression can be correlated with the increase in MDSCs in gastric cancer patients [39]. Similarly, arginase 2 (ARG2) expression has been reported to suppress the proliferation of CD4+ effector T cells and facilitate Treg accumulation by inhibiting the mTOR signaling pathway in metastatic melanoma [37].

Furthermore, the cellular metabolic interaction consists of the disruption of lipids inside the TME to inhibit the immune response. For instance, alterations in lipid metabolism have been observed to support the immunosuppressive landscape by enhancing lipid accumulation in Treg cells, thereby increasing their suppressive functions [40]. Moreover, the augmentation of fatty acid oxidation resulted in M2 tumor-associated macrophages (TAM) polarization within the TME [41]. Additionally, it has been observed that an elevation in fatty acid levels inside polymorphonuclear myeloid-derived suppressor cells (PMN-MDSCs) plays a significant role in the modulation of immune responses and is associated with an unfavorable response to immunotherapy. This phenomenon has been identified as a contributing factor to the advancement of tumor growth [42].

Emerging evidence suggests the pivotal role of reactive oxygen species (ROS) in modulating the TME. Cancer cells often exhibit altered redox status, leading to increased production of ROS [43]. In turn, these ROS further exacerbate the immunosuppressive conditions within the TME by damaging immune cells directly or by modulating signaling pathways that enhance the recruitment or function of immunosuppressive, such as Tregs and MDSC. It is reported that ROS produced by cancer cells leads to SUMO-specific protease 3 (SENP3) production within Treg cells. This SENP3 plays a crucial role in controlling the activity of BACH2, a transcription factor important for the immune suppression mediated by Treg cells that contributes to the antitumor immune suppression mediated by Treg cells [44]. The upregulation of ROS production in MDSCs induced the immunological suppression by the elevation of NADPH oxidase (NOX2) expression controlled by STAT3 transcription factor. Additionally, this increase in ROS generation by MDSCs was found to promote immunosuppression in cancer patients by suppressing the responses of T cells and the activity of NK cells [45]. This oxidative stress environment thus represents another metabolic facet through which cancer cells can evade immune surveillance.

The multifaceted metabolic interactions within the TME underscore the complexity of cancer’s evasion from the immune system. Understanding these metabolic pathways opens the door to novel therapeutic strategies to disrupt the metabolic communications between cancer cells and the immune system. By targeting these metabolic pathways, it is possible to reprogram the TME from a state of suppressive to supportive in the context of antitumor immunity, thereby enhancing the efficacy of current immunotherapies and potentially leading to more effective cancer treatments.

## 5. Metabolic Status in Cancer Cells Associated with Antitumor Immune Cell Function

T cells are essential in eliminating pathogens and tumor cells, among many immune cells. In general, naïve T cells can be activated and differentiated into effector T (Teff) cells by recognizing the antigen peptide–MHC–complex with their T cell receptor. Upon activation, Teff cells increase their utilization of aerobic glycolysis to achieve rapid proliferation and expansion, leading to an increase in bioenergetic and anabolic needs. Along with the glycolytic shift, it has been reported that activated T cells also upregulate glutaminolysis and lipid synthesis. In this regard, there is competition between the high glycolytic demand of cancer cells and T cells, and such competition results in nutrient deprivation in TME to impair effector T cell function [6]. Moreover, high expression of GLUT1 in renal cell carcinoma patients was reported to correlate with the low infiltration of CD8+ T cells [46]. In addition to glucose uptake by cancer cells, the increased glutamine consumption in kidney cancer has been reported to suppress the cytotoxic ability of CD8+ T cells through IL-23-dependent Treg activation [47].

Natural killer (NK) cells are recognized as the first line of defense that naturally exerts the cytotoxic ability against cancer cells without prior sensitization. The activation of NK cells can be determined by a balance between the signals from both activating and inhibitory receptors. Along with effector T cell inhibition, it is known that the highly glycolytic pancreatic cancer cells inhibit NK cell-dependent immune surveillance by inhibiting MICA/B expression, which are the ligands for the NK cell-activating receptor NKG2D [48]. NKp46 is another activating receptor of NK cells, and the lactate from Pan02 pancreatic cancer cells has been reported to downregulate the expression of NKp46 and inhibit antitumor NK cell activity in vivo [49]. In the same context, high LDHA expression and lactate production of melanoma cells were reported to inhibit the survival and antitumor function of both T cells and NK cells [50], and Notch1 also induced glycolytic lactate production in lung cancer cells to inhibit NK cell activation [51]. Additionally, IDO from thyroid cancer cells has been reported to suppress NK cell cytotoxicity by inhibiting NKG2D and NKp46 expression via STAT1 and STAT3 signaling [52].

Dendritic cells (DCs) are important in controlling tumor immunity as professional antigen-presenting cells link innate and adaptive immune responses. It is also known that the metabolic status of cancer cells can induce DC dysfunction and impair antitumor immune response. The TCGA database analysis of head and neck cancer revealed that increased glycolytic status of tumors negatively correlates with the infiltration of DCs to increase tumor immune evasion [53]. Furthermore, tumor-derived lactic acid was reported to alter the antigen presentation of DCs, and GPR81, a G-protein-coupled receptor for lactate, is involved in this process [54,55]. The increased lipid metabolism in cancer cells is also found to impair the antitumor function of DCs [56], whereas the blockade of the mevalonate metabolism in cancer cells induced type 1 classical dendritic cells to facilitate antitumor immunity [57]. Moreover, the activation of FASN was reported to inhibit the function of DCs and initiate the immune escape in ovarian cancer cells by increasing FA production in TME [58].

In summary, metabolic alterations in cancer cells not only support the proliferation and survival of cancer cells but also create a hostile environment for immune cells by depleting essential nutrients and directly inhibiting immune cell functions through various mechanisms. This metabolic competition and direct suppression contribute significantly to tumor immune evasion, complicating the development of effective immune-based cancer therapies.

## 6. Antioxidant Metabolism Associated with Cancer Cell Immune Escape

The high metabolic activity of tumor cells during carcinogenesis significantly increases reactive oxygen species (ROS) level [59]. To survive persistent ROS attack, tumor cells enhance their antioxidant ability (Figure 2). The most abundant cellular antioxidant molecule is glutathione (GSH), and it serves as a cofactor of other enzymes, such as glutathione peroxidases (GPx) or glutathione-S-transferase (GST).

GPx is a selenium-containing antioxidant enzyme that protects cells by reducing hydrogen peroxide and lipid peroxide into water and lipid alcohols. GPx1 expression was reported to be associated with the elevation of immunosuppressive MDSCs and TAM and correlated with worse prognosis in AML patients [60]. The overexpression of GPx2 has been found in the group of immunologically cold-type tumors, inversely correlated with the level of proinflammatory cytokines and associated with poor responsiveness to anti-CTLA-4 therapy [61]. Moreover, there is a report that the high expression of GPx3 in alveolar type 2 (AT2) epithelial cells induced the production of IL-10 in the premetastatic niche. IL-10-producing GPx3 AT2 cells were found to increase Treg and decrease CD4+ T cell proliferation; conversely, the deletion of GPx3 in AT2 cells suppressed lung metastasis [62].

GST is a family of phase II detoxification enzymes that protect cells from the attack of reactive electrophiles. Human GST can be divided into six classes. GST plays a crucial role in regulating the levels of oxidative stress within cancer cells, significantly impacting their growth, advancement, and reaction to therapeutic interventions. As a critical player in cells by detoxifying harmful compounds, GST helps neutralize reactive oxygen species and other carcinogens, thus protecting cells from oxidative damage. GST family enzymes play a critical role in regulating the equilibrium between the production of reactive oxygen species (ROS) and their detoxification. This antioxidant function of GST is particularly important for cancer cells, as their rapid growth and altered metabolism often lead to heightened levels of oxidative stress. Therefore, the increased activity of GST is known to play a critical role in promoting the growth and survival of cancer cells by minimizing oxidative damage to DNA, proteins, and lipids and contributing to the resistance of cancer cells to various therapies. It is known that elevated activity of GST can be a critical factor in supporting the proliferation and survival of cancer cells by reducing oxidative damage to DNA, proteins, and lipids.

Furthermore, GST activity contributes to developing resistance to cancer therapy, posing a significant challenge in treatment strategies [63]. Database analysis of ovarian cancer cells identified that the high expression of GSTM2, -3, and -4 correlates with the decreased expression of immune checkpoint molecules, such as CTLA-4 and TIGIT. Thus, the results highlighted the expression of GSTM2, -3, and -4 in promoting immune escape in ovarian cancer [62]. In addition, GSTA3 was identified as one of the highest contributing features in the study on the genomic and transcriptomic predictors of response to immune checkpoint inhibitors in melanoma patients. The study suggests that GSTA3 plays a significant role in the effectiveness of immune checkpoint inhibitors in treating melanoma and implies the potential of GSTA3 as a genomic biomarker for predicting patient outcomes [64].

In addition, the identification of glutathione-S-transferase-4 (GSTA4) upregulation in melanoma cells that have escaped the immune response marks a significant advancement in the understanding of the ability of cancer cells to evade host immunity. Our recent findings reveal that GSTA4 is crucial in enabling melanoma cells to resist the oxidative stress induced by IFN-γ [65]. This discovery is especially important in melanoma treatment, as there is a pressing need to address resistance to therapies, particularly those that focus on the immune system. This resistance mechanism allows melanoma cells to survive immune attacks and enhances their metastatic capabilities, presenting a dual challenge in cancer treatment. The upregulation of GSTA4 to counteract oxidative stress underscores the adaptability of cancer cells, highlighting the complex interplay between tumors and the host immune system. This balance between immune surveillance and tumor evasion is central to cancer progression, with oxidative stress serving as a pivotal battleground. By increasing their resistance to IFN-γ-induced oxidative stress, melanoma cells survive in an environment rich in immune-mediated stressors and gain a competitive advantage that may facilitate their metastatic spread. This dual benefit to the cancer cells positions GSTA4 as a potential marker for therapeutic targeting. Thus, the study’s findings suggest that inhibiting GSTA4 could disrupt melanoma cells’ defense mechanisms, rendering them more susceptible to immune-mediated destruction and possibly curbing their metastatic capabilities.

Furthermore, the paper highlights the potential of GSTA4 as a biomarker for predicting response to immunotherapy. Given the variability in patient responses to treatments such as immune checkpoint inhibitors, the ability to classify patients based on GSTA4 expression could significantly enhance treatment personalization. The paper also brings to light the broader implications of targeting metabolic pathways in cancer treatment. Focusing on how cancer cells manipulate their metabolic environment to evade immune detection opens new avenues for research into metabolic interventions as cancer therapies. This perspective encourages a holistic view of the tumor microenvironment, considering not just the immune cells and cancer cells but also the complex web of interactions that support tumor growth and spread.

Together with these, tumor cells withstand oxidative stress by manipulating their antioxidant systems and creating an environment that suppresses effective immune responses. This knowledge underscores the potential of targeting these metabolic pathways as a strategy for cancer treatment, opening new avenues for research and therapy development that focus on undermining tumor cells’ metabolic defenses to enhance their susceptibility to immune attack.

## 7. Targeting Cancer Cell Metabolism to Improve Immunotherapy Response

Immune checkpoint inhibitors have dramatically transformed the landscape of cancer therapy by targeting the regulatory pathways that control immune responses against cancer cells. These inhibitors functionally block checkpoint proteins, such as PD-1, PD-L1, and CTLA-4, by which tumor cells enable to evade immune detection. For example, drugs targeting PD-1 or PD-L1 reinvigorate T cells, enhancing their ability to attack cancer cells. CTLA-4 inhibition, meanwhile, boost antitumor immune responses by preventing the negative regulation of T cell activation. This strategic inhibition results in the increased activation of the immune system, specifically enabling cytotoxic T cells to recognize and destroy cancer cells more effectively. The clinical success of these therapies has been remarkable and the survival rates of those who responded to these therapies have significantly improved. Although cancer immunotherapy delivers significant clinical benefits to some patients, many patients do not respond to the therapy or develop resistance to the treatment. Therefore, improving immunotherapy response has emerged, and targeting cancer cell metabolism can be a significant option.

Considering the accumulating evidence of the immunosuppressive role of cancer cell metabolism, its inhibition may improve immunotherapy response. Indeed, the inhibition of LDH with oxamate in combination with pembrolizumab treatment was shown to delay tumor growth and increase the infiltration of activated CD8+ T cells in a preclinical NSCLC model [66]. Moreover, CB-839, the specific glutaminase (GLS1) inhibitor, was reported to improve the efficacy of anti-PD-1 or anti-PD-L1 antibody in a mouse CT26 colon carcinoma model [67]. Such benefit of glutaminase inhibition was also reported in the mouse melanoma model. The combination of CB-839 with immune checkpoint blockade increased IFN-γ-associated T cell inflammatory gene expression [68]. Although the results have not been published yet, the clinical testing of CB-839 in combination with nivolumab has completed a phase I/II study (NCT02771626).

An IDO1 enzyme inhibitor was also shown to improve the efficacy of anti-CTLA-4 and anti-PD-L1 antibody therapy in a mouse B16 melanoma model [69] and further induce the proliferation, survival, and functions of antitumor immune cells in vitro and enhance antitumor immunity in vivo against Pan02 cells [70]. Although the phase I/II clinical trials of epacadostat, a specific IDO1 inhibitor, in combination with immune checkpoint blockade, showed some promise in advanced malignant melanoma patients [71], the phase III clinical study failed to show improvement in progression-free survival or overall survival in metastatic melanoma patients [72]. CB-1158 is an arginase1 inhibitor, and the combination of CB-1158 was shown to improve the efficacy of checkpoint blockade in CT26 and 4T1 mouse tumor models [73]. A clinical study of CB-1158 in combination with pembrolizumab in advanced/metastatic solid tumors is currently underway (NCT02903914).

Statins are a class of drugs that inhibit the production of cholesterol by blocking HMG-CoA reductase and have been reported for their immunomodulatory effects [74,75,76,77]. For instance, simvastatin treatment, tumor antigen vaccination, and anti-PD-1 blockade showed better tumor control and increased survival rates of B16 and TC-1 tumor models [78]. In clinical studies, the combination of statins with immune checkpoint blockade showed better outcomes in metastatic melanoma, pleural mesothelioma, advanced non-small-cell lung cancer patients, and advanced renal carcinoma [79,80,81].

In conclusion, while cancer immunotherapy has significantly benefited many patients, a substantial portion do not respond or develop resistance over time. However, recent advances suggest that targeting cancer cell metabolism offers a promising avenue to enhance the efficacy of immunotherapies. Furthermore, emerging evidence suggests that combining metabolic inhibitors with immune checkpoint blockade can synergistically improve outcomes in cancer treatment. As this research progresses, it holds the potential to expand the therapeutic landscape of immunotherapy, offering hope for improved responses in patients who currently have limited options.

## 8. Future Directions

The interplay between cancer metabolism and immune evasion offers promising research avenues and therapeutic opportunities. As our understanding of this complex relationship deepens, the potential for innovative treatment strategies targeting metabolic and immune pathways will continue to grow. To pursue personalized medicine, developing comprehensive metabolic profiles for individual tumors could predict immune responsiveness and tailor treatments accordingly. Novel technologies, such as metabolomics and proteomics, would provide valuable insights into the metabolic status of cancer cells and their influence on the immune environment [82,83]. Further, advanced imaging and analytical techniques could lead us understand the metabolic gradients within the tumor microenvironment [84]. Nevertheless, our understanding of how metabolic conditions in TME influence immune cell function and tumor progression will be critical for developing new therapeutic strategies. In addition, the heterogeneity of cancer cell metabolism within TME or between individual patients highlights the need for single-cell analysis techniques [85,86,87]. Such higher-resolution analysis of TME could unveil the metabolic signatures correlated with immune evasion, and further guide us to more precise therapeutic strategies.

Identifying key enzymes in metabolic pathways that support immune evasion provides new targets for drug development. Inhibitors of enzymes such as lactate dehydrogenase (LDH) or arginase could disrupt the metabolic support for immunosuppressive cells like myeloid-derived suppressor cells (MDSCs). Moreover, exploring the role of diet and the microbiome in shaping cancer metabolism and the immune landscape might lead to dietary recommendations or probiotic treatments that enhance the effectiveness of cancer therapies.

Adaptive clinical trial designs that can accommodate multiple interventions and biomarker-driven stratification are essential to test the new wave of metabolic–immunological therapies effectively. Rationalizing regulatory pathways for combination therapies and novel metabolic interventions will be critical in efficiently bringing these treatments from bench to bedside.

In conclusion, as the field of cancer research evolves, integrating metabolic and immunological strategies will likely play a pivotal role in the next generation of cancer therapeutics. These approaches aim to disrupt tumors’ metabolic adaptations and strengthen immune cells’ metabolic capabilities, setting the stage for a dual-pronged attack that could significantly improve patient outcomes.

## 9. Concluding Remarks

Cancer cells’ metabolic behavior is critical in facilitating tumor immune escape, primarily through the induction of an immunosuppressive microenvironment and the suppression of effector immune functions (Table 1). The intricate and varied metabolic pathways within tumor cells, reflecting their profound heterogeneity, underscore the challenge of identifying a one-size-fits-all approach to cancer therapy. However, this variability also highlights the importance of thoroughly understanding individual tumor metabolisms to tailor and enhance the effectiveness of immunotherapeutic strategies.

Recent advancements in our understanding of cancer biology reveal that tumor cells can dynamically reprogram their metabolic processes, a survival tactic to evade immunological destruction. This adaptability complicates therapeutic targeting and opens new avenues for intervention. By focusing on the regulatory mechanisms of cancer cell metabolism, researchers and clinicians can develop more precise and potent strategies to disrupt these adaptive cycles, enhancing the body’s ability to combat cancer.

Thus, targeting the metabolic interplay within the tumor microenvironment presents a promising frontier for cancer immunotherapy. As we decipher the complex interactions between cancer metabolism and immune responses, the prospect of devising more effective treatments that can prevent, halt, or reverse tumor progression becomes increasingly feasible. This approach promises to improve therapeutic outcomes and opens the door to potentially curative interventions in oncology.

## Figures and Tables

**Figure 1 ijms-25-05288-f001:**
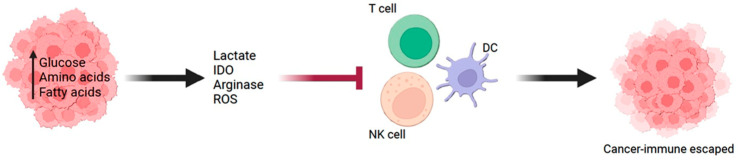
Cancer metabolism and antitumor immunity. Tumor cells have an abundance of nutrient intake to support their metabolism. The metabolic status of cancer cells leads to the immunosuppressive tumor microenvironment that inhibits the effector function of antitumor immune cells.

**Figure 2 ijms-25-05288-f002:**
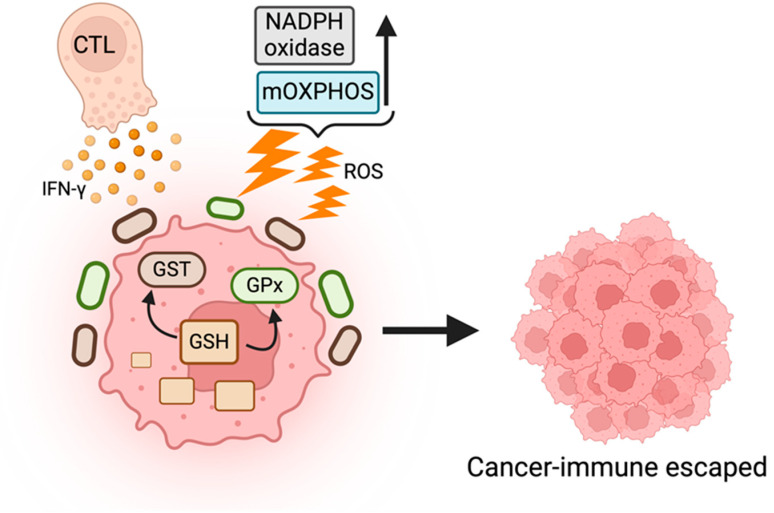
Antioxidant metabolism and cancer cell immune escape. Antioxidant metabolism is involved in the protection of cancer cells from oxidative stress responses. The increase in redox status in cancer cells dampens the attack of immune effector cells and/or activates the immunosuppressive cells, leading cancer cells to escape from immune-surveillance.

**Table 1 ijms-25-05288-t001:** Metabolic changes in cancer cells that control tumor progression.

Molecule	Function	Reference
GLUT	Increase tumor growth and invasion	[8]
HK2	Tumor progression, immune cell infiltration	[10]
SLC1A5	Tumor proliferation and survival	[12]
FASN	Tumor growth and proliferation	[17]
STAT3, c-Myc	Metabolic switch to glycolysis	[19]
Lactate	Activate ERK/STAT3 signaling, activating HIF-1α	[29,30]
IDO	Inhibit effector T cell function, increase Treg recruitment, activating MDSCs	[33,34]
ARG1	Correlation with increase in MDSCs	[39]
ARG2	Suppress CD4+ T cells, increase Treg recruitment	[37]
GPx1	Elevation of MDSCs and TAM	[60]
GPx2	Poor responsiveness to anti-CTLA-4 mAb	[61]
GSTM2, 3, 4	Promoting immune escape	[62]
GSTA4	Resistance to IFN-γ, increase metastatic ability	[65]

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
