# Peer review of "Immunological Aspects of Cancer Cell Metabolism"

_ijms, 2024, doi:10.3390/ijms25105288_

Round 1
Reviewer 1 Report
Comments and Suggestions for Authors
The article titled "Immunological aspects of cancer cell metabolism" presented by Ucche and Hayakawa is a review aiming to describe the relationship between cellular metabolism in tumor cells and its impact on the immune system. I believe the topic is relevant, although I do not clearly understand what novelty the present review contributes to the state of the art. Some of my major comments that require attention to seek a publication site in the IJMS journal are:
-
The review is written in a somewhat disorganized manner within the various sections presented. While coherent writing is appreciated in each section, I do not find a clear thread in the presentation of the different molecules discussed. In this regard, more illustrative diagrams for each section would help summarize the contributions mentioned.
-
A presentation of the tumor immunological components would be appropriate, as well as describing the metabolism of the non-tumor cells in the microenvironment.
-
The present review should be able to provide a new analytical approach or new findings that are not presented in other reviews. I find a lot of similarity with other reviews already published: https://doi.org/10.1038/s41568-020-0273-y; https://doi.org/10.1186/s12943-021-01316-8; https://doi.org/10.1186/s12943-021-01316-8
-
The inclusion of antioxidant systems as part of cellular metabolism needs further justification. It is a response to one of the metabolites (ROS) rather than part of metabolism.
-
The methodology of analysis and article search used to assemble the present review should be described. This would help clarify the focus intended for the manuscript.
- Future perspectives that arise from the analysis provided in the review should be outlined. What does the article aim to show or discuss? The conclusions provided are a simple summary. The discussion of the article as well as the conclusion or future perspectives that other researchers can follow from reading this review should be improved.
Author Response
Comment 1:
The review is written in a somewhat disorganized manner within the various sections presented. While coherent writing is appreciated in each section, I do not find a clear thread in the presentation of the different molecules discussed. In this regard, more illustrative diagrams for each section would help summarize the contributions mentioned.
Response:
We appreciate the valuable suggestions and have tried to improve our writing to make those points clearer in the revised manuscript.
Comment 2:
A presentation of the tumor immunological components would be appropriate, as well as describing the metabolism of the non-tumor cells in the microenvironment.
Response:
In response to the reviewer’s comment, we have added the “immunological components of tumor microenvironment” section in the revised manuscript.
Comment 3:
The present review should be able to provide a new analytical approach or new findings that are not presented in other reviews. I find a lot of similarity with other reviews already published: https://doi.org/10.1038/s41568-020-0273-y; https://doi.org/10.1186/s12943-021-01316-8; https://doi.org/10.1186/s12943-021-01316-8
Response:
Although we understand many relevant review articles have been published elsewhere, we believe our presented review offers a distinct perspective by integrating the latest findings on metabolic interactions within the tumor microenvironment with immunotherapy outcomes which is not yet discussed in the other published articles. In response to the reviewer’s comments, we added more content on recent research outcomes and clinical trials and further proposed novel therapeutic combinations to overcome resistance mechanisms in current immunotherapies in the revised manuscript.
Comment 4:
The inclusion of antioxidant systems as part of cellular metabolism needs further justification. It is a response to one of the metabolites (ROS) rather than part of metabolism.
Response:
We really appreciate the reviewer’s comment. We believe the inclusion of an antioxidant system is very important and would like to clarify why it is essential. First, antioxidant systems are directly linked to metabolic processes because reactive oxygen species (ROS) are byproducts of cellular metabolism and are regulated by cellular antioxidant systems. The regulation of ROS by antioxidant systems is believed to be crucial for maintaining cellular function. In addition, antioxidant systems generally play more roles than clearing ROS. By controlling ROS levels, antioxidant systems maintain the integrity of critical components of cellular metabolism and ensure the efficient function of cellular processes. Therefore, we believe the inclusion of antioxidant systems is relevant and important for our review.
Comment 5:
The methodology of analysis and article search used to assemble the present review should be described. This would help clarify the focus intended for the manuscript.
Response:
We are afraid, but the intention of this comment is somewhat unclear to us. We did not use any particular methodology for writing this review article, but we summarized what we believe are important findings regarding the immunological aspect of cellular metabolism in cancer.
Comment 6:
Future perspectives that arise from the analysis provided in the review should be outlined. What does the article aim to show or discuss? The conclusions provided are a simple summary. The discussion of the article as well as the conclusion or future perspectives that other researchers can follow from reading this review should be improved.
Response:
We thank you for your very constructive comments and now have added the future direction part in the revised manuscript.
Reviewer 2 Report
Comments and Suggestions for Authors
The paper is nicely written and organized. Below I attach some suggestions to improve the paper:
1. Figure 2 is a little generic and does not bring much to the paper. The figure should be more informative.
2. The paper may contain the separate paragraph about the immune checkpoint inhibitors in cancer cells regulation.
3. You may add a paragraph about the role of novel anticancer therapies on the cells metabolism.
4. References should include more recent publications.
Author Response
Comment 1:
Figure 2 is a little generic and does not bring much to the paper. The figure should be more informative.
Response:
We appreciate the valuable comment and now have tried to improve the contents of Figure 2 in the revised manuscript.
Comment 2:
The paper may contain the separate paragraph about the immune checkpoint inhibitors in cancer cells regulation.
Response:
We appreciate the constructive comment and now have added the suggestion in the “Targeting cancer cell metabolism to improve immunotherapy response” section of the revised manuscript.
Comment 3:
You may add a paragraph about the role of novel anticancer therapies on the cells metabolism.
Response:
We thank you for the reviewer’s suggestion but have discussed about the role of novel anti-cancer therapies by the cellular metabolism within the “Targeting cancer cell metabolism to improve immunotherapy response” section. As we would like to specifically discuss the impact of targeting cellular metabolism in cancer immunity, we wish to ask the heading of this section remain as it is.
Comment 4
References should include more recent publications.
Response:
In response to the reviewer’s suggestion, we have now added additional relevant recent publications to the revised manuscript (Ref 9, 18, 82-87).
Reviewer 3 Report
Comments and Suggestions for Authors
Dear Editor,
I carefully reviewed the manuscript entitled “Immunological aspects of cancer cell metabolism”
In this review Authors discuss the potential role of cancer cell metabolism in the context of immune response against cancer cells.
I suggest the suitability for publication pending minor revision since some changes are required in order to improve the quality of the manuscript. I think that English language should be improved throughout the manuscript. I suggest to add a section on new insights on tumor microenvironment and on the small non-coding RNA molecules involved in cell-cell signaling and communication known as MicroRNAs. Indeed, cancer cells commonly induce an intrinsic and chronic inflammatory reaction to stimulate a pro-tumorigenic microenvironment, and the ongoing crosstalk between neoplastic and immune cells results in immunosuppression, invasion and cancer progression. Tumor Microenvironment is the cellular environment in which the tumor exists in the human system. It includes the blood vessels, fibroblast, cells that contribute to the immunity, bone marrow-derived inflammatory cells, lymphocytes, signaling and the extracellular matrix. The tumor existing can interact with the surrounding microenvironment leading to different effects. The tumor can interact with the microenvironment by releasing extracellular signals, can promote tumor angiogenesis and induce peripheral immunity tolerance. The immune cells in the microenvironment can also affect the growth and evolution of the cancerous cells. The tumor microenvironment contributes to the tumor heterogeneity. Tumor microenvironments are recognized as key-contributing factors for studying the cancer progression and drug resistance to the cancer treatment. MicroRNAs (miRNAs), small non-coding RNA molecules involved in cell-cell signaling and communication, have emerged as crucial endogenous regulators of gene expression in cancer. As essential mediators between cancer cells and other cellular components of the tumor microenvironment, miRNAs are considered to be central players in regulating multiple aspects of cancer biology. Please read articles on this field. I can suggest to read and cite the following papers: “Abbate J.M., Arfuso F., Riolo K., Capparucci F., Brunetti B., Lanteri G. EPIGENETICS IN CANINE MAMMARY TUMORS: UPREGULATION OF MIR-18A AND MIR-18B ONCOGENES IS ASSOCIATED WITH DECREASED ERS1 TARGET MRNA EXPRESSION AND ERA IMMUNOEXPRESSION IN HIGHLY PROLIFERATING CARCINOMAS. Animals 13, 2023, 1086.
Abbate J.M., Arfuso F., Riolo K., Giudice E., Brunetti B., Lanteri G. UPREGULATION OF MIR-21 AND PRO-INFLAMMATORY CYTOKINE GENES IL-6 AND TNF-Α IN PROMOTING A PRO-TUMORIGENIC MICROENVIRONMENT IN CANINE MAMMARY CARCINOMAS. Research in Veterinary Science, 2023, 164, 105014.
Abbate J.M., Giannetto, A., Arfuso, F., Brunetti, B., Lanteri, G. RT-qPCR EXPRESSION PROFILES OF SELECTED ONCOGENIC AND ONCOSUPPRESSOR MIRNAS IN FORMALIN-FIXED, PARAFFIN-EMBEDDED CANINE MAMMARY TUMORS. Animals 12, 2022, 2898.”
Moreover, Authors should indicate the inclusion/exclusion criteria used for the choice of the papers.
Comments on the Quality of English LanguageEnglish language should be improved throughout the manuscript.
Author Response
Comment 1:
In this review Authors discuss the potential role of cancer cell metabolism in the context of immune response against cancer cells.
I suggest the suitability for publication pending minor revision since some changes are required in order to improve the quality of the manuscript. I think that English language should be improved throughout the manuscript.
Response:
Thank you very much for the suggestion. We will discuss with the editorial team whether we need to receive the language editing.
Comment 2:
I suggest to add a section on new insights on tumor microenvironment and on the small non-coding RNA molecules involved in cell-cell signaling and communication known as MicroRNAs. Indeed, cancer cells commonly induce an intrinsic and chronic inflammatory reaction to stimulate a pro-tumorigenic microenvironment, and the ongoing crosstalk between neoplastic and immune cells results in immunosuppression, invasion and cancer progression. Tumor Microenvironment is the cellular environment in which the tumor exists in the human system. It includes the blood vessels, fibroblast, cells that contribute to the immunity, bone marrow-derived inflammatory cells, lymphocytes, signaling and the extracellular matrix. The tumor existing can interact with the surrounding microenvironment leading to different effects. The tumor can interact with the microenvironment by releasing extracellular signals, can promote tumor angiogenesis and induce peripheral immunity tolerance. The immune cells in the microenvironment can also affect the growth and evolution of the cancerous cells. The tumor microenvironment contributes to the tumor heterogeneity. Tumor microenvironments are recognized as key-contributing factors for studying the cancer progression and drug resistance to the cancer treatment. MicroRNAs (miRNAs), small non-coding RNA molecules involved in cell-cell signaling and communication, have emerged as crucial endogenous regulators of gene expression in cancer. As essential mediators between cancer cells and other cellular components of the tumor microenvironment, miRNAs are considered to be central players in regulating multiple aspects of cancer biology. Please read articles on this field. I can suggest to read and cite the following papers: “Abbate J.M., Arfuso F., Riolo K., Capparucci F., Brunetti B., Lanteri G. EPIGENETICS IN CANINE MAMMARY TUMORS: UPREGULATION OF MIR-18A AND MIR-18B ONCOGENES IS ASSOCIATED WITH DECREASED ERS1 TARGET MRNA EXPRESSION AND ERA IMMUNOEXPRESSION IN HIGHLY PROLIFERATING CARCINOMAS. Animals 13, 2023, 1086.
Abbate J.M., Arfuso F., Riolo K., Giudice E., Brunetti B., Lanteri G. UPREGULATION OF MIR-21 AND PRO-INFLAMMATORY CYTOKINE GENES IL-6 AND TNF-Α IN PROMOTING A PRO-TUMORIGENIC MICROENVIRONMENT IN CANINE MAMMARY CARCINOMAS. Research in Veterinary Science, 2023, 164, 105014.
Abbate J.M., Giannetto, A., Arfuso, F., Brunetti, B., Lanteri, G. RT-qPCR EXPRESSION PROFILES OF SELECTED ONCOGENIC AND ONCOSUPPRESSOR MIRNAS IN FORMALIN-FIXED, PARAFFIN-EMBEDDED CANINE MAMMARY TUMORS. Animals 12, 2022, 2898.”
Moreover, Authors should indicate the inclusion/exclusion criteria used for the choice of the papers.
Response:
We totally agree with those valuable suggestions from the reviewer regarding the importance of small non-coding RNAs in regulating tumor microenvironments. Although we do understand such importance, we wish to focus the present review to summarize how cellular metabolism affect cancer cell biology and immunity. We have added more content in response to other reviewers' comments to improve the quality of our review.
Reviewer 4 Report
Comments and Suggestions for Authors
This review/perspective focuses on a very important topic - the role of metabolism in the fight between immune and tumor cells. This has been extensively studied primarily in the tumor cells, but a growing number of studies look at the metabolism within the immune cells. The authors in this article highlight an interesting angle to the metabolic regulation of the tumor microenvironment. I was surprised to learn that c-Myc expression increased through a IFN-STAT3-dependent manner, and found that section of the article interesting. Additionally I thought the "Targeting cancer cell metabolism to improve immunotherapy response" was interesting and beneficial to the reader. I think this manuscript could be improved with some editing to make certain points more succinct. There are several large paragraphs which could be shortened or broken into smaller paragraphs. Below are some suggestions.
Suggestions:
-PAGE 1, LINE 17 “Normal healthy cells can evolve into transformed cancer cells through various mechanisms” is vague. Maybe list 1-3 mechanisms briefly.
-PAGE 1, The first paragraph is very long. I would suggest making a break in Line 30
-PAGE 2, Line 54 “only when oxygen is scarce is glucose fermented to lactic acid.” The Warburg Effect occurs at high oxygen levels. Although hypoxia will prevent oxidative phosphorylation from occurring, driving fermentation; hypoxia is not a requirement for cancer cells to ferment glucose during high rates of proliferation. This statement should be edited.
-PAGE 2, Lines 57-58 “Through this glycolytic metabolic reprogramming, cancer cells rapidly produce ATP than the conventional OXPHOS pathway.” Needs to be reworded. Perhaps “Cancer cells rapidly produce ATP through this glycolytic metabolic reprogramming rather than the conventional OXPHOS pathway.”
-PAGE 2 is a single paragraph. This should be broken into smaller paragraphs for the reader.
-PAGE 5 LINES 190-191 “Teff cells increase bioenergetic and anabolic needs to achieve rapid proliferation and expansion, thereby shifting their metabolism from OXPHOS to aerobic glycolysis.” Teff cells do not shift their metabolism from OXPHOS to aerobic glycolysis. Both OXPHOS and aerobic glycolysis are actually increased following T cell activation; however, there is an increase in the percentage of the ATP generated from aerobic glycolysis post activation. I would restate “Teff cells increase bioenergetic and anabolic needs to achieve rapid proliferation and expansion by increasing their utilization of aerobic glycolysis”
-Individual sections have abrupt endings rather than summary/conclusion of the section
Comments on the Quality of English LanguagePlease see suggestions above.
Author Response
Comment 1:
PAGE 1, LINE 17 “Normal healthy cells can evolve into transformed cancer cells through various mechanisms” is vague. Maybe list 1-3 mechanisms briefly.
Response:
In response to the reviewer’s suggestion, we have described several general mechanisms for cancer cell transformation accordingly in the revised manuscript.
Comment 2:
PAGE 1, The first paragraph is very long. I would suggest making a break in Line 30
Response:
In response to the reviewer’s comment, we have inserted breaks and made the first paragraph into several sentences.
Comment 3:
PAGE 2, Line 54 “only when oxygen is scarce is glucose fermented to lactic acid.” The Warburg Effect occurs at high oxygen levels. Although hypoxia will prevent oxidative phosphorylation from occurring, driving fermentation; hypoxia is not a requirement for cancer cells to ferment glucose during high rates of proliferation. This statement should be edited.
Response:
We appreciate the reviewer’s suggestion and now have edited the statement accordingly in the revised manuscript.
Comment 4:
PAGE 2, Lines 57-58 “Through this glycolytic metabolic reprogramming, cancer cells rapidly produce ATP than the conventional OXPHOS pathway.” Needs to be reworded. Perhaps “Cancer cells rapidly produce ATP through this glycolytic metabolic reprogramming rather than the conventional OXPHOS pathway.”
Response:
We reworded that sentence accordingly in response to the reviewer’s suggestion.
Comment 5:
PAGE 2 is a single paragraph. This should be broken into smaller paragraphs for the reader.
Response:
In response to the reviewer’s comment, we have made the content of page 2 into smaller paragraphs.
Comment 6:
PAGE 5 LINES 190-191 “Teff cells increase bioenergetic and anabolic needs to achieve rapid proliferation and expansion, thereby shifting their metabolism from OXPHOS to aerobic glycolysis.” Teff cells do not shift their metabolism from OXPHOS to aerobic glycolysis. Both OXPHOS and aerobic glycolysis are actually increased following T cell activation; however, there is an increase in the percentage of the ATP generated from aerobic glycolysis post activation. I would restate “Teff cells increase bioenergetic and anabolic needs to achieve rapid proliferation and expansion by increasing their utilization of aerobic glycolysis”
Response:
We appreciate the suggestion and now have changed it accordingly in the revised manuscript.
Comment 7:
Individual sections have abrupt endings rather than summary/conclusion of the section
Response:
Thank you very much for the suggestion. We have now added the conclusive statements in each section where relevant.
Round 2
Reviewer 1 Report
Comments and Suggestions for Authors
I consider that the authors did an interesting editing job and added valuable information to the review. I have no further comments.
Author Response
Thank you very much for your valuable comments.
Reviewer 2 Report
Comments and Suggestions for Authors
The Authors adressed all my concerns, however, some errors still have to be corrected: 1. The abstract is too short and not informative - please make a teaser of your work which would attract others to read it and cite it. 2. References in table 1 should also be in brackets as these are just like other references.
Author Response
We regret those mistakes and now revised the manuscript accordingly as
Comment 1
The abstract is too short and not informative - please make a teaser of your work which would attract others to read it and cite it.
Response:
We have revised the abstract in response to the reviewer's suggestion.
Comment 2
References in table 1 should also be in brackets as these are just like other references.
Response: We have adjusted the reference style and number in Table 1 in response to the reviewer's comment.
Reviewer 4 Report
Comments and Suggestions for Authors
The manuscript has been improved.
Author Response

(The authors gave the same response as above.)
